# Bilateral Asymmetries Assessment in Elite and Sub-Elite Male Futsal Players

**DOI:** 10.3390/ijerph17093169

**Published:** 2020-05-02

**Authors:** Jorge López-Fernández, Jorge García-Unanue, Javier Sánchez-Sánchez, Enrique Colino, Enrique Hernando, Leonor Gallardo

**Affiliations:** 1Centre for Sport, Exercise and Life Science, Coventry University, Coventry CV1 5FB, UK; 2IGOID Research Group, University of Castilla-La Mancha, Toledo, 45071 Castilla-La Mancha, Spain; jorge.garciaunanue@uclm.es (J.G.-U.); Enrique.Colino@uclm.es (E.C.); Enrique.Hernando@uclm.es (E.H.); Leonor.Gallardo@uclm.es (L.G.); 3School of Sport Science, Universidad Europea, Villaviciosa de Odón, 28670 Madrid, Spain; javier.sanchez2@universidadeuropea.es

**Keywords:** equilibrium, leg asymmetry, performance, physiology, sport

## Abstract

This study aimed to investigate morphological, functional, and neuromuscular asymmetries on futsal players’ lower limbs at different competitive levels. Sixteen male elite futsal players from the Spanish National Futsal League and thirteen male sub-elite futsal players from the third division participated in this study. Morphological asymmetry was assessed through bioelectrical impedance (fat-mass (g and %) and lean-mass (g)). Functional asymmetry was assessed by means of a 20-s static unipedal balance test. Finally, neuromuscular asymmetry was assessed using tensiomiography tests on both the rectus femoris (RF) and biceps femoris (BF) of each participant. The three tests conducted did not reveal significant bilateral asymmetries in elite players. On the other hand, sub-elite players showed significant bilateral asymmetry in fat-mass percentage between dominant and non-dominant limbs (+6%; CI95%: 1 to 11; ES: 0.88; *p* = 0.019). They also showed higher bilateral asymmetry in the delay time of the RF (+13%; CI95%: 7 to 21; ES: 1.3; *p* < 0.05). However, the static unipedal balance test (*p* > 0.05) did not evidence asymmetries regardless of the level of the participants. Elite futsal players do not develop bilateral asymmetries in lower limbs in the studied parameters. On the contrary, sub-elite players are likely to develop morphological and neuromuscular asymmetries between their dominant leg and non-dominant leg.

## 1. Introduction

It is well acknowledged that most injuries in futsal occur in the lower limbs [1,2,3,4,5,6]. For instance, in the Spanish national futsal team 31.1% of all injuries affect right-lower limb, 29.0% left-lower limb and 24.6 affect both right- and left-lower limbs at the same time [4]. Most of them are due to contusion, tendon rupture and muscular damage [1,2,3,4,5,6]. Therefore, gaining knowledge about the status of futsal player’s lower limbs might help to identify the likelihood of suffering injuries and to prescribe individualized exercises to reduce it [7,8]. In this regard, the bilateral asymmetries (deviations from mirror symmetry between the dominant leg and the non-dominant leg) [8] could contribute to this purpose as they have been related to different musculoskeletal injuries [9,10,11]. Furthermore, the development of these asymmetries has also been related to a detriment of performance [8,12,13]. Thus, improving our understanding about the lower limb asymmetries in futsal players might contribute to reducing the likelihood of getting injured and the related performance loss [8,9,10,11,12,13].

In futsal, like in soccer, players adopt unstable positions when performing the technical actions and sports gestures (i.e., kicks, ball control, dribbles, jumps, tackles, accelerations, decelerations, etc.) [14,15]. Therefore, players’ ability to stabilize the posture when performing these actions is essential in order to avoid injuries [16,17], but also to get the most under a performance perspective [8,18,19]. However, it is well acknowledged that players of these sports frequently perform the technical actions like kicking or controlling the ball with the dominant limb while the non-dominant limb is more often used to stabilize the body [15]. Thus, this role difference might contribute to the likelihood of development inter-limb asymmetries [15], and may explain why most of the injuries occur in the dominant lower limb [1,2,4]. In this regard, the evidence suggests that lower level’s players may be more likely to develop greater inter-limb asymmetries due to their lesser ability in using both legs to perform technical actions and the less intense training regime [13,20,21].

Currently there are several instruments and protocols (e.g., tools, wearables, tests, indexes) susceptible of being used for identifying morphological, functional, muscular, and neuromuscular asymmetries [7,22,23]. Among them, those related to the ability of the player to stabilize the posture play a key role in the injury risk and performance as players have to deal with external factors like contusions, pushes, collisions, etc. while performing the technical actions at a high intensity [17,18,21,24,25]. Moreover, both local and whole-body fatigue reduce players’ postural control due to alterations in the sensory information from the proprioceptive system [26]. This increases the postural sway (constant, slight corrective deviations from vertical when standing upright) [5] and therefore the likelihood of suffering a sports injury [16,27]. The ability to maintain equilibrium or balance on one leg is characterized by the muscular synergies that reduce the movement of the center of pressure (COP) when players perform a static or dynamic motor action on a single-leg stance [8,14,16]. Higher balance ability is associated with a reduced movement of the COP. When assessing the postural control and the balance ability of athletes by recording variations in the COP [14,16,20], it is important to consider that these factors vary from sport to sport and from person to person [14,28], with soccer, and thus futsal, being among the most demanding sports [29,30]. Several researchers suggest that this is because players of this sports frequently have to hold themselves on one leg while the other is used to perform technical actions like kicks, ball control, dribbles, etc. [15,29,30,31]. However, as said, this also may lead players to develop inter-limb asymmetries in their ability to hold equilibrium, with higher stability in the non-dominant limb than the dominant one [20]. Moreover, as lower level players have a less intense training regime and in general have lower ability to perform the technical actions with the non-dominant limb, it is likely that sub-elite futsal players show greater stability when holding on their non-dominant limb than their elite futsal counterparts [32].

On the other hand, the ability to hold balance with one leg is related to variables such as lean mass, muscular tone, or muscular force [8,29,33]. Therefore, the information provided by the single-leg stance tests may be complemented by assessing futsal players’ lower limbs composition and contractile properties of their muscles. Moreover, under a performance perspective, thigh and shank lean mass asymmetry account for up to 20% of the variance in propulsive force [34], while more accurate kickers show lower lean-mass asymmetry [13]. In this case, previous studies evidence differences in body composition according to the playing level [13,35], which could lead to the development of different levels of asymmetries between dominant and non-dominant lower limbs’ tissue composition [13,36]. To measure composition of different body segments and assess potential morphology asymmetries, dual energy X-ray absorptiometry (DXA) is the most accurate instrument [13]. However, DXA is a very expensive technique and requires players to visit a specific lab where the test needs to be conducted for several minutes. To address this issue, several authors have opted for the bioelectrical impedance method because this equipment is much more affordable, can be easily transported and the test only lasts for a few minutes [37,38]. This instrument has proven to have sufficient inter- and intra-reliability to be used for both scientific and sport purposes and to provide information about morphological asymmetries in tissue composition of players’ extremities [37,38]. This is due to the ability of the equipment to distinguish parameters such as total fat mass or lean mass either in kg or percentage (%) on different body segments (right- and left-upper limbs, right- and left-lower limbs, and torso).

On the other hand, previous studies suggest that bilateral asymmetries can be identified by measuring the speed of contraction on symmetric muscles using a determined electric stimulus [39,40,41]. This is performed by means of tensiomyography (TMG), a non-invasive technique that does not affect performance for subsequent training sessions, allows individual muscle assessment and allows a large number of athletes to be measured in a short period of time [42]. Even though there are some controversies regarding the reliability and reproducibility of TMG [43], this tool has shown a high interclass correlation for measuring maximal muscular displacement (Dm; ICC: from 0.94 to 0.99), contraction time (Tc; ICC: from 0.92 to 0.98), and delay time (Td; ICC: from 0.86 to 0.97) in lower-limb muscles [44,45,46,47]. In fact, it has been used to assess bilateral asymmetries in elite futsal players’ lower limbs [48], but there are no studies analyzing these types of asymmetries in futsal players of different competitive levels.

As stated before, the electrical bioimpedance, the single leg stance tests, and the TMG can be used to identify three different types of bilateral asymmetries in futsal players’ lower limbs, and these asymmetries may vary depending on players level. However, to the authors’ knowledge, no one has previously used these three techniques in futsal players of different competitive levels. For that reason, the objective of this research was to use these tests to study morphological, functional, and neuromuscular asymmetries in futsal players’ lower limbs in different competitive levels.

## 2. Methods

### 2.1. Participants

Sixteen male futsal players from three elite clubs belonging to the Spanish National Futsal League (LNFS) (25.8 ± 5.8 years; 176.2 ± 5.3 cm; 74.85 ± 5.17 kg; 12 right-footed; 4 left-footed) and thirteen sub-elite male futsal players from two clubs belonging to the third division (23.2 ± 4.62 years; 173 ± 6 cm; 71.25 ± 6.33 kg; 10 right-footed; 3 left-footed) participated in this study. Goalkeepers were not included in the study.

The study protocol was approved by the Local Ethics Committee according to the Code of Ethics of the World Medical Association (Declaration of Helsinki). In addition to an agreement with the clubs, all participants were informed that participation in the study was voluntary. Before conducting the study, participants were informed about the objectives and characteristics of this study, the test procedure, and the associated risks. Finally, all participants were requested to sign the informed consent and get approval from the medical department of their club.

### 2.2. Procedure

Participants were grouped according to their playing level (elite (n = 16) and sub-elite (n = 13) players). The study was conducted during the competitive period, within a short break of the national competition. Each club arranged three days for the data collection process. On the first day, participants performed an initial pilot test to familiarize themselves with the tests included in the study protocol. These tests did not include any exhaustive physical activity. During the second day, participants agreed not to perform any exhaustive physical activity to allow for a recovery time of 48 h. Finally, the testing session was conducted on the third day in the indoor training field of each club (average temperature of 23 ± 3 °C). During the study, all participants were instructed to maintain their usual nutritional habits.

### 2.3. Experimental Protocol

#### 2.3.1. Tissue Composition of Lower Limbs

The composition of both legs (fat mass (g and %) and lean mass (g)) of participants were measured by bioelectrical impedance (Tanita BC418-MA, Tanita Corp., Tokyo, Japan) [37]. Participants were asked to not to eat 3 h prior to the test, they were encouraged to drink around 30 mL of water 90 min prior to the study in order to be hydrated when conducting the test. Additionally, players were asked to urinate a few minutes before the test.

#### 2.3.2. Postural Sway Test

After the tissue composition assessment, players were asked to perform a static unipedal balance test for 20 s, either with the dominant leg or the non-dominant one [16]. This test was conducted on a force plate (Kistler 9286BA, Kistler Group, Eulachstrasse, Switzerland), and players were instructed to be as static as possible during the test. To consider the test valid, the arms had to be held in the lumbar region while the non-supporting leg was raised so that the limb had a knee flexion of 90 °C and the suspended foot was approximately at the supporting limb’s malleolus height [16]. The recovery time between trials (dominant leg–non-dominant leg) as well as the recovery time in invalid tests was set at 30 s. The starting supporting leg was randomly assigned for each participant so that half of the participants of each group (elite players and sub-elite players) started the test with the dominant leg, and the other half with the non-dominant leg.

Postural sway data were recorded using the manufactured software (Mars Balance and Stability, Kistler Group, Eulachstrasse, Switzerland), registering the following variables: total sway path (mm); anteroposterior sway path (mm); mediolateral sway path (mm); total sway speed (mm/s); anteroposterior sway speed (mm); mediolateral sway speed (mm); anteroposterior sway average amplitude (mm); mediolateral sway average amplitude (mm); anteroposterior sway maximal amplitude (mm); mediolateral sway maximal amplitude (mm); total sway area (mm^2^); anteroposterior sway area (mm^2^); mediolateral sway area (mm^2^); total sway area per second (mm^2^/S); anteroposterior sway area per second (mm^2^/S); mediolateral sway area per second (mm^2^/S); anteroposterior free from peaks (Hz); mediolateral free from peaks (Hz).

#### 2.3.3. Contractile Properties of Lower-Limb Muscles

The contractile properties of the rectus femoris (RF) and biceps femoris (BF) in basal condition were recorded by measuring the response of these muscles to an induced electric stimulus using a TMG equipment (TMG-100 System electrostimulator, TMG-BMC d.o.o., Ljubljana, Slovenia). The electric stimulus was provoked by two self-adhesive electrodes (TMG electrodes, Ljubljana, Slovenia), and the muscle response was measured with a digital Dc-Dc transducer Trans-Tek^®^ (GK 40, Ljubliana, Slovenia) placed perpendicular to the muscle belly and equidistant from the self-adhesive electrodes at a distance of 50–60 mm.

All measurements were performed by the same expert technician, applying six electrical stimuli for 1ms (10, 25, 50, 75, 100, and 110 mA). Fifteen seconds of recovery between measurements were allowed to minimize the effects of potentiation and fatigue. The RF was assessed with participants in supine position with a 120° knee flexion, while the BF was evaluated with participants in prone position and the knee flexed to 5° [46]. Participants did not report any discomfort during this test. The variables assessed in this study were the maximum radial displacement of the muscle belly (Dm), contraction time (Tc), and delay time (Td) because these parameters have been proven to have a low error level (0.5% to 2.0%) and a high reproductivity (ICC: 0.85–0.98) [39,45,46,47].

### 2.4. Statistical Analysis

Data are displayed as mean ± SD. The normality of the distribution of variables and homogeneity of variance were checked using the Kolmogorov–Smirnov test and Levene’s statistic. The differences between competitive level (elite vs. sub-elite) and bilateral asymmetry (dominant leg vs. non-dominant leg) were analyzed by two-way ANOVA and Bonferroni post-hoc test. The differences in the bilateral asymmetry between elite and sub-elite were analyzed with an independent sample t-test. The bilateral asymmetry was defined as the difference between both dominant and non-dominant leg ((dominant leg–non-dominant leg)/dominant leg). Participant’s age and weight were used as cofactors when performing post-hoc analysis. Confidence interval of the differences (CI of 95%) was included and effect size was calculated to identify the magnitude of changes. The ES was evaluated according to the following criteria: 0 to 0.2 = trivial, 0.2 to 0.6 = small, 0.6 to 1.2 = moderate, 1.2 to 2.0 very large, 2.0 to 4.0 nearly perfect and >4.0 = perfect [49]. SPSS 21.0 was used for the data analysis and the level of significance was established at *p* < 0.05.

## 3. Results

The comparative assessment between elite and sub-elite players showed a similar fat mass percentage in both groups (*p* > 0.05). However, elite players had higher lean mass than sub-elite both in the dominant (+0.54 kg; CI95%: 0.10 to 0.98; ES: 0.89; *p* = 0.017) and non-dominant (+0.61 kg; CI95%: 0.17 to 1.05; ES: 1.03; *p* = 0.007; Figure 1) legs. Finally, sub-elite futsal players showed higher bilateral asymmetry in fat mass percentage than elite futsal players (+6%; CI95%: 1 to 11; ES: 0.88; *p* = 0.019).

Regarding the static unipedal balance test (Table 1), the outcomes did not reveal significant bilateral asymmetries in the participants, regardless of their playing level (*p* < 0.05). Although sub-elite players had higher medio-lateral sway area (+12.11 mm·s; CI95%: 1.73 to 22.48; ES: 0.78; *p* < 0.05; Table 2) and higher medio-lateral sway area per second (+0.60 mm; CI95%: 0.09 to 1.12; ES: 0.78; *p* < 0.05), they did not show higher bilateral asymmetries than elite players (*p* < 0.05).

Results from the TMG are displayed in Table 2. Regarding the RF, sub-elite players showed higher Td values in the non-dominant leg than elite players (+2.59 ms; CI95%: 0.71 to 4.46; ES: 0.91; *p* < 0.05) and significant bilateral symmetry (+13%; CI95%: 7 to 21; ES: 1.3; *p* < 0.05). On the other hand, sub-elite players also demonstrated higher Td values in BF than elite players either in the dominant leg (+2.22 ms; CI95%: 0.71 to 3.74; ES: 0.97; *p* < 0.05) or the non-dominant leg (+1.79 ms; CI95%: 0.29 to 3.29; ES: 0.90; *p* < 0.05).

## 4. Discussion

This research assessed morphological, functional, and neuromuscular asymmetries in lower-limb muscles of futsal players in different competitive levels by evaluating tissue composition, single-leg static balance, and contractile properties of RF and BF, respectively. The main findings are that elite players did not present asymmetries in any of the three tests conducted, whilst sub-elite players evidenced significant bilateral asymmetries in lower limb’s fat mass (morphological asymmetry) and in Td of RF (neuromuscular asymmetry). On the other hand, when comparing elite versus sub-elite players, the former showed lower sway area in the dominant leg and higher lean mass on both legs than the latter, but none of the groups displayed significant asymmetries in these parameters.

The importance of measuring either the tissue composition of athlete’s whole body or just of a specific corporal segment is well acknowledged, as a lower percentage of fat mass and higher lean mass are related to higher performance [13,35]. To the authors’ knowledge, there are no reference values of tissue composition of lower limbs for elite futsal players or even elite soccer players by using bioimpedance. However, a previous study assessing body composition in Premier League soccer players suggests that lower limbs lean mass of elite players oscillate between 10.23 kg and 13.16 kg [36]. These parameters concur with those of the elite players in our study, but not with the sub-elite ones as some of them presented values lower than 10.23 kg. Regarding bilateral asymmetries, previous studies have evidenced that lower playing level is related to higher morphological asymmetries in player’s lower limbs [13,36]. The findings of the present study are in line with these works as sub-elite players showed higher asymmetries in fat mass percentage. However, although sub-elite players presented lower lean mass compared to elite players, no asymmetries (*p* > 0.05) were found for this variable. This may be due to the low sample of sub-elite players included in this study (n = 13) as it has been theorized that elite players develop more balanced lower limbs to cope with the increasing demands of elite matches [50]. Moreover, the higher training regime of elite players might lead them to having a higher lean mass in lower limbs than their sub-elite counterparts [35].

Regarding the imbalance ability of futsal players, neither sub-elite players nor elite displayed bilateral asymmetries when performing the static unipedal balance test for 20 s. These findings are in line with those of Matsuda et al. [30], who found that soccer players do not have asymmetries between their dominant and non-dominant leg. However, they performed a 60-s unipedal test instead of 20-s, which could somehow affect the results [30,51]. In this regard, shorter tests conducted in soccer players (5-s and 10-s tests) suggest that there is higher stability in the non-dominant leg than in the dominant one [14,20], probably due to the non-dominant generally being more used to stabilize the body posture when controlling, dribbling, or kicking [15]. Therefore, the outcomes presented in this manuscript are somewhat surprising as it was expected that futsal players had higher stability in the non-dominant leg than in the dominant one [14,15], which could explain why contact and overuse injuries predominantly occur in the dominant leg [4,52]. In any case, due to the low sample size of this study, care must be taken when considering the findings of this work. Moreover, the findings from Kartal [14] and Barone et al. [20] suggest that tests longer than 10 s might conceal bilateral asymmetries in futsal and soccer players maybe due to the fact that futsal actions usually last for less than 5 s [53]. Thus, further research is required to confirm if both elite and sub-elite futsal players have a different imbalance ability between the dominant and non-dominant lower limb and the differences between these two groups. In this last aspect, elite players’ dominant leg showed a lower relative and absolute sway area in the anterior–posterior axis than sub-elite players, which somehow might support the hypothesis that elite players have greater stability than sub-elite players [18] because they spend a greater number of hours in practice than sub-elite ones [14].

Regarding the outcomes from the TMG, both elite and sub-elite players showed a great ability to rapidly generate force during contractions (Tc and Td lower than 30 ms) and have high muscle tone, as reflect their low Dm values [39,54]. These findings are quite similar to those found in soccer players during competition [54,55], although each group presented some differences for each variable. These differences may be due to soccer and futsal having different specific demands (i.e., futsal is more dynamic than soccer and is played on a smaller court). However, further studies should test this hypothesis because Dm, Td, and Tc values might change throughout the competitive season [39,55,56]. Regarding the competitive level, sub-elite futsal players showed higher Td values in RF (non-dominant leg) and BF (dominant and non-dominant leg) than elite ones which may suggest that elite futsal players are more explosive than sub-elite and they have a greater number of type 1 fibers [45,57]. However, caution is required because no differences between the playing level were found in Tc [39,45]. In the case of asymmetries, elite players did not present bilateral asymmetries neither in RF nor in BF. On the contrary, sub-elite players displayed significantly greater Td in the RF of the non-dominant leg, but no other asymmetries were found. Although clear conclusions cannot be taken from these results, it is likely that these differences are because the dominant leg is frequently used to produce rapid moments when kicking the ball, while the non-dominant leg is mainly used for body support [15]. Future works inducing fatigue in participants might provide further information about whether sub-elite futsal players have asymmetries in RF between the dominant and non-dominant leg [48].

A limitation of the present study could be the size of the sample, as only 16 elite players and 13 sub-elite players were tested. This study was conducted during a short break of the national competition so once this time was over it was not possible to access to a higher sample. In this sense, future studies should attempt to increase the sample size to improve the power of the outcome and its interpretation. Another limitation of the study might be the use of the TMG, since there is no current consensus about the reliability and reproducibility of this equipment [43]. However, this tool has shown a high interclass correlation for Dm (ICC: from 0.94 to 0.99), Tc (ICC: from 0.92 to 0.98), Td (ICC: from 0.86 to 0.97), and Ts (ICC: from 0.79 to 0.96) in lower-limb muscles [44,45,46,47]. Moreover, the co-activation of neighboring muscles might influence the outcomes of the test [43].

## 5. Conclusions

Elite futsal players are not predisposed to developing bilateral asymmetries in the tissue composition of lower limbs, single-leg equilibrium ability, and contractile properties of the RF and BF. Sub-elite players may develop bilateral asymmetries in the tissue composition of lower limbs with greater fat mass in the non-dominant leg and in the Td of RF. Although sub-elite players have lower-lean mass in lower limbs and lower mediolateral sway axis balance stability, no asymmetries were found for these parameters.

## Figures and Tables

**Figure 1 ijerph-17-03169-f001:**
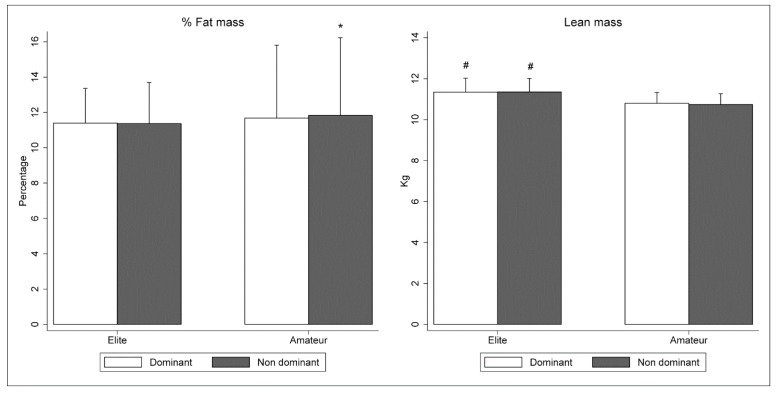
Fat mass percentage (%) and lean mass (kg) of both dominant leg and non-dominant leg of elite and sub-elite futsal players.^#^ Significant differences between elite and sub-elite futsal players (*p* < 0.05); * Significant differences between dominant and non-dominant lower limb.

**Table 1 ijerph-17-03169-t001:** Static unipedal balance test performance of both dominant and non-dominant leg in elite and sub-elite futsal players.

	Dominant Leg		Non-Dominant Leg		Bilateral Asymmetry	
	Elite Players	Sub-Elite Players	Elite Players	Sub-Elite Players	Elite Players	Sub-Elite Players
Sway path-total (mm)	819.92 ± 207.44	880.77 ± 232.17	828.56 ± 222.46	844.31 ± 215.76	−0.03 ± 0.19	0.03 ± 0.16
Sway path-A-P (mm)	507.88 ± 155.60	577.46 ± 180.25	522.52 ± 131.04	555.46 ± 160.87	−0.07 ± 0.26	0.02 ± 0.17
Sway path-M-L (mm)	542.80 ± 138.76	551.69 ± 134.72	545.92 ± 169.14	526.08 ± 144.59	−0.02 ± 0.17	0.04 ± 0.19
Sway V-total (mm/s)	40.97 ± 10.34	44.08 ± 11.68	41.46 ± 11.15	42.19 ± 10.77	−0.03 ± 0.19	0.03 ± 0.16
Sway V-A-P (mm/s)	25.40 ± 7.79	28.88 ± 9.00	26.13 ± 6.56	27.78 ± 8.04	−0.07 ± 0.26	0.02 ± 0.17
Sway V-M-L (mm/s)	27.14 ± 6.95	27.59 ± 6.75	27.31 ± 8.45	26.30 ± 7.24	−0.02 ± 0.17	0.04 ± 0.19
Sway average amplitude-A-P (mm)	5.54 ± 1.91	6.28 ± 2.30	6.27 ± 2.04	5.84 ± 1.69	−0.18 ± 0.41	0.01 ± 0.26
Sway average amplitude-M-L (mm)	6.34 ± 1.89	6.29 ± 2.12	6.28 ± 1.92	5.88 ± 2.37	0.00 ± 0.20	0.06 ± 0.23
Sway maximal amplitude-A-P (mm)	33.91 ± 7.02	35.15 ± 7.01	33.84 ± 8.98	36.40 ± 8.10	−0.08 ± 0.42	−0.07 ± 0.30
Sway maximal amplitude-M-L (mm)	25.36 ± 4.09	28.31 ± 5.08	25.03 ± 5.13	26.58 ± 4.24	−0.01 ± 0.28	0.04 ± 0.18
Sway area-total (mm^2^)	2082.96 ± 782.15	2312.31 ± 855.88	2032.80 ± 660.08	2112.31 ± 639.90	−0.08 ± 0.45	0.04 ± 0.28
Sway area-A-P (mm·s)	106.52 ± 27.25	114.59 ± 39.68	104.92 ± 28.48	107.41 ± 30.94	−0.03 ± 0.38	0.00 ± 0.34
Sway area-M-L (mm·s)	79.79 ± 13.37 ^#^	91.90 ± 17.41	80.17 ± 16.52	83.62 ± 13.54	0.00 ± 0.24	0.07 ± 0.20
Sway area per second-total (mm^2^/s)	104.10 ± 38.98	115.79 ± 42.96	101.75 ± 33.08	105.68 ± 31.98	−0.08 ± 0.45	0.04 ± 0.28
Sway area per second-A-P (mm)	5.33 ± 1.36	5.73 ± 1.98	5.25 ± 1.43	5.37 ± 1.55	−0.03 ± 0.38	0.00 ± 0.34
Sway area per second-M-L (mm)	3.99 ± 0.67 ^#^	4.59 ± 0.86	4.01 ± 0.83	4.18 ± 0.67	0.00 ± 0.24	0.07 ± 0.19
FRE from peaks-A-P (Hz)	4.69 ± 0.60	4.86 ± 1.31	4.36 ± 0.94	4.87 ± 0.98	0.06 ± 0.14	−0.04 ± 0.25
FRE from peaks-M-L (Hz)	4.41 ± 0.68	4.52 ± 0.61	4.47 ± 0.80	4.70 ± 0.71	−0.03 ± 0.08	−0.04 ± 0.10

^#^ Significant differences between elite and sub-elite futsal players (*p* < 0.05); * Significant differences between dominant leg and non-dominant leg (*p* < 0.05). A-P: Anteroposterior; M-L: Mediolateral.

**Table 2 ijerph-17-03169-t002:** Tensiomyography (TMG) values of rectus femoris and biceps femoris of dominant leg and non-dominant leg in elite and sub-elite futsal players.

		Dominant Leg		Non-Dominant Leg		Bilateral Asymmetry	
		Elite Players	Sub-Elite Players	Elite Players	Sub-Elite Players	Elite Players	Sub-Elite Players
RF	Td (ms)	23.92 ± 2.24	23.21 ± 2.75	23.44 ± 1.78	26.03 ± 3.91 *^,#^	0.02 ± 0.09	−0.12 ± 0.11 ^#^
	Tc (ms)	30.96 ± 8.37	30.99 ± 5.95	28.34 ± 6.06	31.40 ± 7.99	0.06 ± 0.18	−0.02 ± 0.17
	Dm (mm)	6.86 ± 2.50	7.21 ± 2.07	7.43 ± 2.95	6.32 ± 2.83	−0.10 ± 0.43	0.10 ± 0.35
BF	Td (ms)	23.18 ± 1.62	25.40 ± 2.98 ^#^	22.37 ± 1.87	24.16 ± 2.10 ^#^	0.03 ± 0.11	0.04 ± 0.09
	Tc (ms)	26.33 ± 5.90	33.96 ± 12.97	30.60 ± 14.59	33.27 ± 10.03	−0.21 ± 0.68	−0.09 ± 0.47
	Dm (mm)	5.84 ± 2.10	5.97 ± 2.12	5.56 ± 2.79	5.93 ± 2.52	0.00 ± 0.47	−0.02 ± 0.36

^#^ Significant differences between elite and sub-elite fustal players (*p* < 0.05); * Significant differences between dominant leg and non-dominant leg (*p* < 0.05); Td: delay time; Tc: contraction time; Dm: maximal muscular displacement.; RF: Rectus femoris; BF: Biceps femoris.

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
