# Peer review of "Bilateral Asymmetries Assessment in Elite and Sub-Elite Male Futsal Players"

_ijerph, 2020, doi:10.3390/ijerph17093169_

Round 1

Reviewer 1 Report

The authors provided a study on the physiological characteristics of futsal players.  The study is sound, and I would recommend publishing it. However, the paper needs to be edited for various formatting and grammatical errors.  

  1. The authors should indicate that these are male players.  I would recommend that the word male be inserted in the title between "elite" and "futsal." Also, in line 98, insert male between sixteen and futsal. 
  2. Line 12, place a comma after injury.
  3. Line 14, place a comma after asymmetry.
  4. Line 30, delete "Whilst," as it seems to be unnecessary.
  5. Line 32, insert "the fact that" after despite
  6. Line 37, add e.g., after the parenthesis, and delete the etc.)
  7. Line 40, "Among of all them" is awkwardly written.  Rewrite to be more clear. 
  8. Lines 47-49, These lines are written awkwardly. The transition from the previous sentence seems too abrupt, so can this be rewritten?
  9. Line 53, "leg seems might be" is worded oddly.  This needs to be clarified.
  10. Line 57, hyphenate single-leg stance.  Multi-word descriptors should be hyphenated before the noun they are modifying. This should be done throughout the manuscript for several items, e.g., lower-limb muscles, higher-balance ability.
  11. Line 58, replace sportsmen with athletes. This is not a big deal, but athletes would be a better word in this instance.  Often sportsmen carries with it the connotation of hunting and things of that nature, and this study is dealing with competitive athletes playing futsal.
  12. Line 75, change need to needs to be grammatical correct.
  13. Line 78, should inter- and intra-reliability be used here?
  14. Line 94, place a comma after asymmetry
  15. Line 109, change they to the participants or the subjects or a similar term.
  16. Line 112, insert "The" in front of Participants and use a lowercase p. This is a minor issue, but using the article "the" is better English.
  17. Line 128, add a comma after Switzerland). The two sentences are independent statements and need to be separated by a comma. 
  18. Line 151, add a comma after Slovenia). 
  19. Line 156, change was to were. The subject (15 sec) is plural.
  20. Line 164, change was to were, again this is for subject-verb agreement.
  21. Line 169, should it be "The participants' ages and weights were..."?
  22. Lines 176, 180, should it be fat-mass percentage?
  23. Same issue in the legend of Fig. 1.
  24. Lines 186 and 187, the * and # statements should be single spaced and be listed within the legend of the figure.
  25. In Table 1, the data points do not need to go to a second line, e.g., 207.44 should be on the same line. Please adjust the Table. This happens numerous times throughout the table.
  26. The tables in general are very crowded and difficult to read at times.  I understand why this is, but perhaps they can be reworked some to make them easier to read.  Table 2 has unnecessary underscoring occurring. Also, the noted statements with * and # need to be closer to the bottom of the Table.  In fact, I would prefer to see them be made as part of the table, if possible.
  27. Line 210, should it be assessed instead of assesses?  This is a formatting issue that the editor should decide. Technically the study has already occurred, so past tense may be used.  However, some editors prefer present tense be used as much as possible.
  28. Line 221, well acknowledged needs to be hyphenated when it precedes the word it is modifying. In this instance, it comes after so the hyphen is not necessary.
  29. Line 228, replace "are probably due to" with "may be because of"
  30. Line 231, delete "up"; it is unnecessary. 
  31. Line 236, the phrase "this study evidence" is odd.  Can this be rewritten to be clearer?
  32. Line 248, add a comma after work.
  33. Line 297, delete the comma after players.

Author Response

We would like to thank the feedback provided by Reviewer 1. We have carefully addressed all the suggested improvements accordingly. Please see the attached document.

Reviewer 2 Report

The manuscript suffers from several concerns that preclude it from a possible acceptance as in the present form. These are listed below.

Overall

  • The manuscript must be proofread. Several English language grammar and syntax errors arose from the manuscript. I didn’t want to list all sentence-by-sentence changes that should be done, so please check the whole manuscript.

Abstract

  • The abstract is written using two different fonts. Please amend.
  • From the abstract, I have no idea of what the Authors really did. Please rewrite it, highlighting the methods and the results, maybe deleting the long background.

Introduction

  • The introduction is overall poor and does not lead me towards the aim of the study. More in detail,
  • 1st paragraph: There is not any mention of the type of injury and its prevalence.
  • 2nd paragraph: “factors” is inappropriate, please reword. This paragraph is redundant and unclear, so I do not understand why it is useful in this form. Please rewrite it.
  • 3rd paragraph: “adequate”, “correct” can be interpreted according to personal opinion. Please reword in more transparent definitions. Additionally, several speculations are argued, so there is no description of any possible direct mechanisms.
  • 4th paragraph: the possible relationship between the balance ability and the inter-limb asymmetry (in what??) must be clearly explained.
  • 5th paragraph: again, why the listed devices could be useful? And specifically, which parameters?
  • 6th paragraph: as such, no rationale is provided.

Methods

  • Participants: why the goalkeepers were excluded from the study?
  • Procedures: please provide the study design and a calculation of the sample size.
  • Tissue composition: please provide more details.
  • Statistical analysis: I suggest using the more conservative Hopkins’ recommendations to interpret the effect size.

Discussion

  • The discussion is overall poor and does suffer from the poorly explained rationale within the introduction.

Author Response

We would like to thank the feedback provided by Reviewer 2. We have carefully addressed all the suggested improvements accordingly. Please see the attached document.

Round 2

Reviewer 2 Report

The Authors have overall improved the manuscript. However, some concerns still remain.

  • The introduction contains several paragraphs, albeit each paragraph does not deal with a specific topic. I suggest merging the paragraphs dealing with each specific topic for clarity.
  • The manuscript does not examine the relationship between the dependent parameters and the injuries occurrence in futsal, rather examine the asymmetry in the dependent parameters in two different futsal populations. Therefore, both in the introduction and discussion, the Authors should examine and discuss this aspect. The possible relationship with injuries must be clearly addressed (when necessary) and it must be clear that these are speculations.
  • Asymmetries are not only possibly related to injury but are also detrimental for performance. Additionally, the Authors should mention both inter-limb and anterior-posterior asymmetries. Please check Coratella et al. 2018, Hum Mov Sci and Coratella et al. 2015 J Sport Sci and Coratella et al 2018 Sport Sci Health.
  • Please give a further check to the English language. A few errors are still present.

Author Response

The Authors have overall improved the manuscript. However, some concerns still remain.

Once again, we appreciate the suggestions provided for Reviewer 2 and acknowledge its contribution to the improvement of this manuscript.

The introduction contains several paragraphs, albeit each paragraph does not deal with a specific topic. I suggest merging the paragraphs dealing with each specific topic for clarity.

We have merged the paragraphs as requested. Thank you very much

The manuscript does not examine the relationship between the dependent parameters and the injuries occurrence in futsal, rather examine the asymmetry in the dependent parameters in two different futsal populations. Therefore, both in the introduction and discussion, the Authors should examine and discuss this aspect. The possible relationship with injuries must be clearly addressed (when necessary) and it must be clear that these are speculations.

We have tried our best to make the requested changes either in the introduction or the discussion. We sincerely appreciate this feedback

Asymmetries are not only possibly related to injury but are also detrimental for performance. Additionally, the Authors should mention both inter-limb and anterior-posterior asymmetries. Please check Coratella et al. 2018, Hum Mov Sci and Coratella et al. 2015 J Sport Sci and Coratella et al 2018 Sport Sci Health.

We have followed this instruction in the implemented changes, thank you very much. In this case, we decided not to mention anterior-posterior asymmetries because we do not study this type of asymmetries. In this study, we assessed femoris and biceps femoris trough the TMG but we only focused on bilateral asymmetries and we did not analyse agonist-antagonist asymmetry between these two muscle groups.

Please give a further check to the English language. A few errors are still present.

An English native has reviewed the latest version